# Effect of pH Variations on the Yield Stress of Calcium Bentonite Slurry Treated with pH-Responsive Polymer

**DOI:** 10.3390/ma13112525

**Published:** 2020-06-01

**Authors:** Hyunwook Choo, Youngmin Choi, Woojin Lee, Changho Lee

**Affiliations:** 1Department of Civil Engineering, Kyung Hee University, Yongin 17104, Korea; choohw@khu.ac.kr; 2Infra Engineering Team 2, SK Engineering & Construction Co., Ltd., Seoul 04534, Korea; choiym@sk.com; 3School of Civil, Environmental, and Architectural Engineering, Korea University, Seoul 02841, Korea; woojin@korea.ac.kr; 4Department of Civil Engineering, Chonnam National University, Gwangju 61186, Korea

**Keywords:** polyacrylamide, calcium bentonite, yield stress, pH, zeta potential

## Abstract

The pH-responsive polymers, such as polyacrylamide (PAM), show distinct conformational states according to the pH of their environmental groundwater. Therefore, the interactions between clay–polymer and polymer–water molecules, which determine the yield stress of bentonite–polymer composites, can be affected by the pH of groundwater. This study aims to evaluate the effect of pH variation on the yield stress of calcium bentonite treated with PAM. The yield stresses (*τ_y_*) of untreated and PAM treated clays were measured with varying volume fractions of solid (VF = 10–23%) and under varying pH conditions (pH = 7.6–9.6). In addition, the zeta potential was measured for both untreated and treated clays to figure out the change in the surface charge of the mineral surface due to PAM treatment. The results of this study demonstrate that *τ_y_* for treated clay is higher than that for untreated clay at a given VF, because van der Walls attraction dominates electrostatic repulsion in the case of treated clay. Due to the change in conformational states of PAM and the consequent change in surface charge that comes with varying pH, the pH-dependent change in *τ_y_* of treated clay is significantly different from that of untreated clay.

## 1. Introduction

Bentonite slurry has been widely used for various engineering purposes, such as slurry walls, grouts, and boring fluids, due to its low hydraulic conductivity and high swelling characteristics. Although sodium bentonite is widely used due to its superior engineering performance, natural deposits of calcium bentonite are globally more common than those of sodium bentonite [1,2]. Due to the calcium ion, calcium bentonites show relatively poor engineering performance compared to sodium bentonites [1,3,4,5,6,7]. Therefore, to satisfy engineering requirements, a low-quality calcium bentonite is typically admixed with additives, such as phosphate [8,9,10], sodium carbonate [11], and polymer [3,12].

When the bentonite slurry is employed as a grout material, its rheological properties, such as yield stress and viscosity, are very important. Especially, yield stress is the key parameter determining the injectability of grout into the ground and also determining resistivity against erosion by groundwater flow after installing the grout [13,14,15]. Note that yield stress is defined as “the stress above which the material flows like a viscous fluid” [16]. It is well known that the yield stress of bentonite slurry can be affected by a number of parameters, including the clay concentration, type of bentonite, molar ratio of Na/Ca, microstructure, electrolyte concentration, and pH [9,11,17,18,19,20].

If bentonite slurry is admixed with a polymer, the factors influencing yield stress become more complex due to the additional interactions between clay–polymer and polymer-pore fluid [3]. Previous studies on clay–polymer composite have focused on the type of polymer and polymer concentration in order to explore interactions between clay–polymer and polymer–water molecules [21,22,23,24]. In contrast, studies evaluating the effect of pH on the yield stress of clay–polymer composite are limited, although the interactions between clay–polymer and polymer–water molecules can be changed by the pH of groundwater. Especially, pH-responsive polymers, such as polyacrylamide (PAM), show distinct conformational states according to the pH of environmental groundwater. Therefore, to estimate the effect of pH on the yield stress of clay–polymer composite, in this study, a calcium bentonite was treated with a nonionic PAM, and the yield stresses of both untreated and treated clay were measured according to the various clay contents and pH conditions. In addition, zeta potential values of untreated and treated clay were measured to gain an insight into the surface charge of the mineral surface.

## 2. Materials and Methods

### 2.1. Materials

#### 2.1.1. Bentonite

In this study, a calcium bentonite produced by Donghae Chemicals Industrial (Korea) was used as a base material. The clay was sieved through a No. 200 sieve to minimize large-sized impurities. The mineralogy of the bentonite was measured using an X-ray diffractometer (XPERT MPD, Philips, Almelo, the Netherlands, maximumradiation; 3kW) as shown in Figure 1. The angle scanned was 2°–42° (2θ) at a rate of 0.04°/s (2θ/s). The main clay mineral was revealed to be a montmorillonite. The chemical composition of the bentonite (Table 1) was measured using X-ray fluorescence (XRF) spectrometry (Philips, Almelo, the Netherlands, PW2404, 60 kV; 125 mA; maximum of 4 kW for an X-ray tube with a Rh anode). Since the Na^+^/Ca^2+^ ratio (0.53) was smaller than 1, the bentonite used in this study can be classified as a calcium bentonite. The index properties of the tested calcium bentonite are given in Table 2. The specific gravity (*G_s_*) was determined to be 2.51 by water pycnometer method [25]. The plastic limit (PL) was determined to be 35.83% by the traditional thread-rolling method [26], and the liquid limit (LL) was determined to be 86.76% by the fall cone method [27]. Therefore, the tested calcium bentonite was classified as a clay with high plasticity (CH), in accordance with the Unified Soil Classification System [28]. The specific surface area (*S_a_*) was 260.56 m^2^/g according to the methylene blue spot test [29], and the pH value was 8.48 for 2% solid contents. The cation exchange capacity (CEC) was measured to be 89.65 meq/100g by the ammonium acetate method [30].

#### 2.1.2. Polyacrylamide (PAM)

In this study, a nonionic PAM (Yangfloc N-100P, OCI-SNF, Seoul, Korea) was chosen to create a PAM treated clay because of its large deformable capacity with varying pH [31]. As shown in Figure 2a, the nonionic PAM contracts at a low pH and extends at high pH. Therefore, it was expected that the clay–PAM composite can have pH-dependent engineering properties. The molecular weight and charge density of the PAM were 8.5 × 10^6^ g/mol and −0.56–1.23 meq/100 g (data from the manufacturer), respectively. Interaction between PAM molecules and clay particles can be achieved by: (1) the carbonyl oxygen (‒C=O) of PAM molecules can form hydrogen bonds between PAM and the clay surface (Figure 2b); and (2) ion–dipole interactions can occur between the polar group (‒NH_2_) of PAM molecules and the interlayer cations of clay (Figure 2b) [32]. It is through these reactions that the clay–PAM composite is formed.

### 2.2. Preparation of PAM Treated Clay

The PAM treated clay was prepared following the method suggested by [34]. 300 g of the dried clay was mechanically blended with 6 L of deionized water for 24 h to ensure complete dispersion. 1 L of the nonionic PAM solution was then poured into the slurry. The concentration of PAM in the slurry was 1.88 g/L (131.6 mg of the PAM for 100 g of the clay). The concentration of 1.88 g/L was chosen, because the swelling index of the PAM treated clay, determined by [35], reaches a maximum value (≈7.4 mL/2 g) at the concentration of 1.88 g/L. After continuous stirring for 24 h, the slurry was allowed to settle down for 1 h. Any supernatant liquid was drained out, then the slurry was dried in an oven at 110 °C for one day. Prior to performing any experiments, the dried clay was ground with a mortar and pestle, then sieved through a No. 200 sieve. The index properties of the PAM treated clay are tabulated in Table 2.

### 2.3. Yield Stress Measurement

Yield stress of the slurry specimen was measured using a rheometer (Brookfield, RST-SST, Middleborough, MA, USA) with a 4 blades vane spindle (VT-80-70; height of vane = 80 mm and width of a vane blade = 35 mm). The 1.1 L slurry specimen was prepared in a 1.2 L tall-form beaker (inner diameter 93 mm), then the slurry was rested for 24 h to complete the hydration reaction. After the equilibration time, the slurry was thoroughly mixed for 1 min, then the yield stress was measured at the spindle rotational speed of 0.5 rpm. After the yield stress measurement, the pH of the specimen was measured by a pH strip (Advantec, Dublin, CA, USA), and the supernatant liquid was extracted by centrifuging from a small sample to measure the electrical conductivity of pore water in the specimen using a conductivity meter (Fisher Scientific, XL50, Pittsburgh, PA, USA).

## 3. Results and Discussions

### 3.1. Effect of PAM Treatment on Stress–Time Curve

Figure 3 shows typical stress–time curves of untreated and treated clays at respective water contents (*w* = weight of water/weight of solid) of 218% and 263%. Figure 3a shows a typical stress–time curve for the untreated sample: shear stress linearly increases with time initially before reaching a maximum value, it then decreases to a quasi-constant value. Previous studies have defined the peak stress in a stress–time curve as the yield stress (*τ_y_*) [36,37,38,39]. However, for the PAM treated specimen, the peak stress was not observed in the stress–time curve under the tested solid contents and pH conditions (Figure 3b). [40] also observed the absence of a peak stress for PAM treated kaolin clay in their results from a consolidated undrained triaxial test. Therefore, the *τ_y_* of the treated clay was determined as the shear stress, at which the stress initiated the post-peak hardening behavior, as illustrated in Figure 3b.

The difference in the stress–time profiles between the untreated and treated clays corresponds to differences in the microstructures of the slurries. In the case of the untreated clay slurry, an aggregated or dispersed structure developed due to the calcium ions [1,3,5]. As the untreated specimen is sheared by the vane, the aggregated/dispersed structure is destroyed at the peak shear strength, followed by a reduction in stress due to slips and rearrangements between clay particles. However, in the case of the treated clay slurry, the adsorbed PAM on the clay surface resulted from hydrogen bonds and ion-dipole interactions between PAM and the clay particles. However, in the case of the treated clay slurry, the adsorbed PAM on the clay surface, resulted from hydrogen bonds and ion-dipole interactions between PAM and the clay particles, can develop the molecular bridges between adjoining clay particles, known as polymer bridging [34,40,41,42]. This results in the prevention of slips and rearrangements between particles after reaching yield stress, and gives rise to the post-peak hardening behavior characteristic of PAM treated clay.

### 3.2. Effect of Solid Content on the Yield Stress

Yield stress (*τ_y_*) is significantly affected by the solid content of the sample [9,43,44]. To explore the effect of PAM treatment on the yield stress of the tested calcium bentonite, the yield stresses of untreated and treated clays with varying solid contents were measured. Figure 4 shows *τ_y_* of untreated and treated clays as a function of volume fraction of solid (VF) at a pH of around 8.4, which is the natural pH without adding any pH adjusting agent, measured by a pH strip. Note the natural pH of pore fluids obtained by centrifuging samples was also determined to be 8.4–8.5, measured by a pH meter (Fisher Scientific, XL50, Pittsburgh, PA, USA). The VF was calculated according to:(1)$$VF=\frac{V_{solids}}{V_{total}}=\frac{1}{\left(1+w\cdot G_{s}\right)}$$
where *V_soilds_* is volume of solid, *V_total_* is total volume, *w* is water content, and *G_s_* is specific gravity. As shown in Figure 4, *τ_y_* for the treated clay was higher than that for the untreated clay at a given *VF*, reflecting the clear impact of PAM on the rheological behavior of the tested bentonite. It can also be observed in Figure 4, that *τ_y_* of both untreated and treated clays exponentially increases with increasing *VF*; thus, *τ_y_* of both clays are proportional to VF in a semi-logarithmic plot.

The network structure of clay particles (i.e., the card-house structure, resulting from edge (+)/face (−) contacts; and the band-like structure, resulting from face (−)/face (−) contacts), which determines *τ_y_* of clay slurry, is affected by surface charge of the clay particle surface [20]. As zeta potential gives an insight into the surface charge of a mineral surface, many previous studies have investigated the direct relationship between the zeta potential and yield stress of clay slurry [9,45]. Figure 5 shows the zeta potential (*ζ*) of the tested untreated and treated clays as a function of pH, ranging from 2.8 to 11.8. At a pH of around 8.4, it can be observed in Figure 5 that the absolute *ζ* of treated clay is smaller than that of untreated clay, because the nonionic PAM adsorbed on the clay particle surface neutralizes the surface charge of clay particles [46]. In the case of treated clay, this results in the van der Walls attractive force overwhelming the electrostatic repulsive force, resulting from the diffuse double layer with counter-ions. Consequently, the treated clay can form the dense structure. Therefore, the yield stress of treated clay at a given solid content is greater than that of untreated clay.

Additionally, because the adsorbed PAM provides polymer bridging between particles [34,40,41], the polymer bridging may act as flexible reinforcement and may provide additional shear resistance by increasing the internal kinematic constraints on soil particles. This explanation of the contribution of polymer bridging to the increase in *τ_y_* can be further supported by comparing the slopes between *τ_y_* and *VF* of treated and untreated clays (Figure 4): the difference in *τ_y_* between treated and untreated clays increases with *VF,* because the chance to form the polymer bridging between particles also increases.

### 3.3. Effect of pH Variations on the Yield Stress

Figure 6a shows variation of yield stress (*τ_y_*) as a function of pH for the treated and untreated clays at volume fractions of solid (*VF*), of 11.5% and 15.3%, respectively. Note that the selected *VF* values for the two tested materials are based on the similar *τ_y_* at a pH of 8.4 (natural pH without adding pH adjusting agent), as shown in Figure 6a. Figure 6a clearly demonstrates that *τ_y_* values of both untreated and treated clays increases as pH decreases. The microstructure of bentonite clay tends to form a flocculated structure (i.e., increasing edge (+)/face (−) interaction) with decreasing pH [18,20]. As the strength of a flocculated structure is higher than that of a dispersed structure [9,20,47], an increase in *τ_y_* with decreasing pH is expected. In contrast, when the pH increases, the microstructure of bentonite clay tends to form a dispersed structure (i.e., increasing face (−)/face (−) interaction), resulting in a decrease in *τ_y_*.

The pH adjustment (i.e., either decreasing pH by addition of HNO3 or increasing pH by addition of NaOH) accompanies an increase in the electrolyte concentration within pore fluid. This increase in the electrolyte concentration can lead to an increase in *τ_y_* due to a reduction in repulsive force between clay particles [7,20] and the consequent formation of a denser structure. Therefore, an increase in *τ_y_* with changing pH can be accelerated in low pH conditions, due to the combined effects of pH and electrolyte concentration (Figure 6a). In contrast, the increase in *τ_y_* with changing pH is very gradual or minimal in high pH conditions, due to the offset mechanism between the pH and electrolyte concentration (Figure 6a): an increase in OH^-^ concentration leads to the expansion of the diffuse double layer and the consequent increase in repulsive force between particles; at the same time, high electrolyte concentration leads to a decrease in repulsive force.

In order to directly compare pH-dependent changes in *τ_y_* of untreated and treated clays, a normalized *τ_y_*, which is defined as the ratio between *τ_y_* and *τ_y_* at a pH of 8.4 (natural pH), is plotted as a function of pH in Figure 6b. Figure 6b demonstrates that as pH decreases from 8.4 to lower values, the treated clay shows a bigger increase of the normalized *τ_y_* than untreated clay. In contrast, as the pH increases from 8.4 to higher values, the normalized *τ_y_* shows very gradual increase in the case of untreated clay; while, the treated clay shows a very slight decrease in the normalized *τ_y_*. This difference in the pH-dependent change of normalized *τ_y_* for untreated and treated clays reflects that the varying conformational states of PAM with pH have a significant influence on *τ_y_* of the tested calcium bentonite. As the PAM molecules contract with decreasing pH (Figure 2), the resulting microstructure of treated clay becomes denser than untreated clay because of the decreased repulsive force between clay particles, which resulted from the increased clay surface coverage by the highly coiled PAM. The increase in the difference between the zeta potential (*ζ*) values of untreated and treated clays with decreasing pH from 8.4 to lower values may support this explanation (Figure 5). Therefore, the bentonite treated with PAM shows a greater increase in yield stress with decreasing pH than untreated bentonite. In contrast, as the PAM molecules extend with increasing pH (Figure 2), the resulting microstructure of treated clay was in a loose (dispersed) state, due to the neutralization of the surface charge by the adsorbed PAM, and was reduced due to a decrease in the region of attached segments of polymer on the clay surface. The *ζ* values of treated clay approach those of untreated clay at high pH in Figure 5 and also supports this explanation. Furthermore, the net change in *ζ,* when pH increases from pH 8.4 to higher values of treated bentonite, is greater than that of untreated bentonite. Therefore, the treated clay may experience significant increases in repulsive forces between particles when changing pH from 8.4 to higher values. Consequently, it can be postulated that, in the case of treated clay, the increase in repulsive force due to the pH change is greater than the decrease in repulsive force due to the increase of electrolyte concentration. This results in the treated clay showing a slight decrease in yield stress when increasing pH from 8.4 to higher values; while, untreated clay shows a gradual increase in yield stress, as shown in Figure 6.

## 4. Summary and Conclusions

This study measured the yield stresses (*τ_y_*) of untreated and PAM treated clays, with varying volume fractions of solid (VF = 10–23%) and varying pH conditions (pH = 7.6–9.6). The key findings of this study are:(1)The peak stress was not observed in the stress–time curve of the treated bentonite, because the polymer bridging between clay particles can prevent slips and rearrangements between particles after reaching *τ_y_*.(2)*τ_y_* for the treated clay was higher than that for untreated clay at a given *VF*, because van der Walls attraction dominates electrostatic repulsion in the case of treated clay. In addition, polymer bridging may act as flexible reinforcement and may provide additional shear resistance, leading to increases in *τ_y_* for the treated clay.(3)An increase in *τ_y_* with decreasing pH can be accelerated in low pH conditions due to the combined effects of pH (i.e., formation of flocculated structure) and electrolyte concentration (i.e., formation of dense structure).(4)The varying conformational states of PAM with pH significantly influences the *τ_y_* of tested calcium bentonite. Therefore, the treated clay shows greater increases in normalized *τ_y_* than untreated clay as the pH decreases from 8.4 to lower values.

## Figures and Tables

**Figure 1 materials-13-02525-f001:**
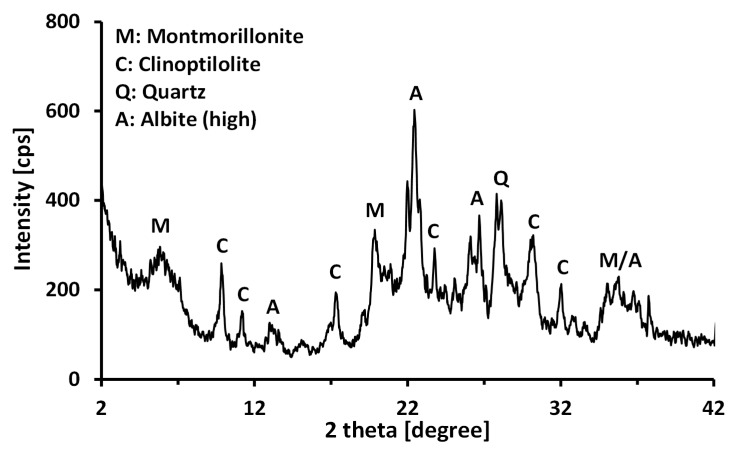
X-ray diffraction (XRD) pattern of the calcium bentonite used in this study.

**Figure 2 materials-13-02525-f002:**
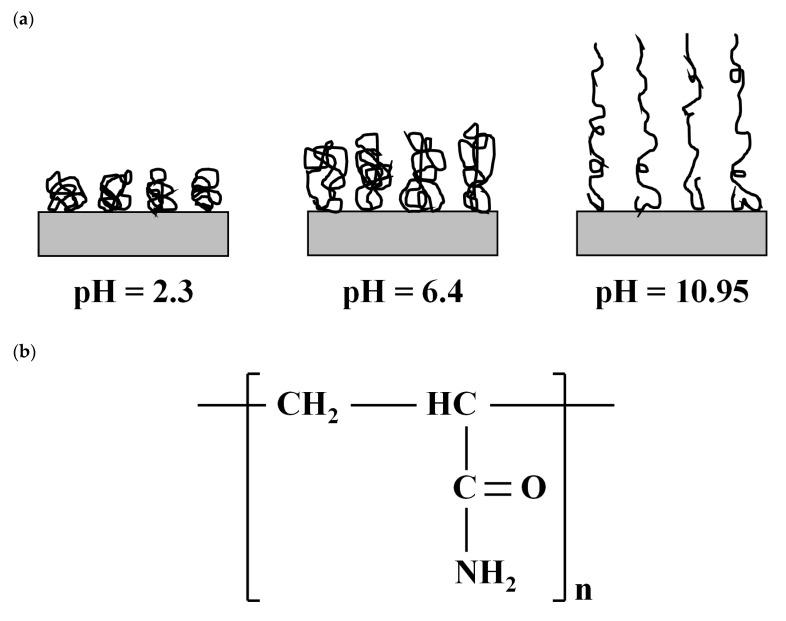
Nonionic polyacrylamide (PAM): (**a**) schematic drawing of the conformational state of adsorbed PAM at different pH conditions (after [31]) and (**b**) chemical structure of PAM (after [33]).

**Figure 3 materials-13-02525-f003:**
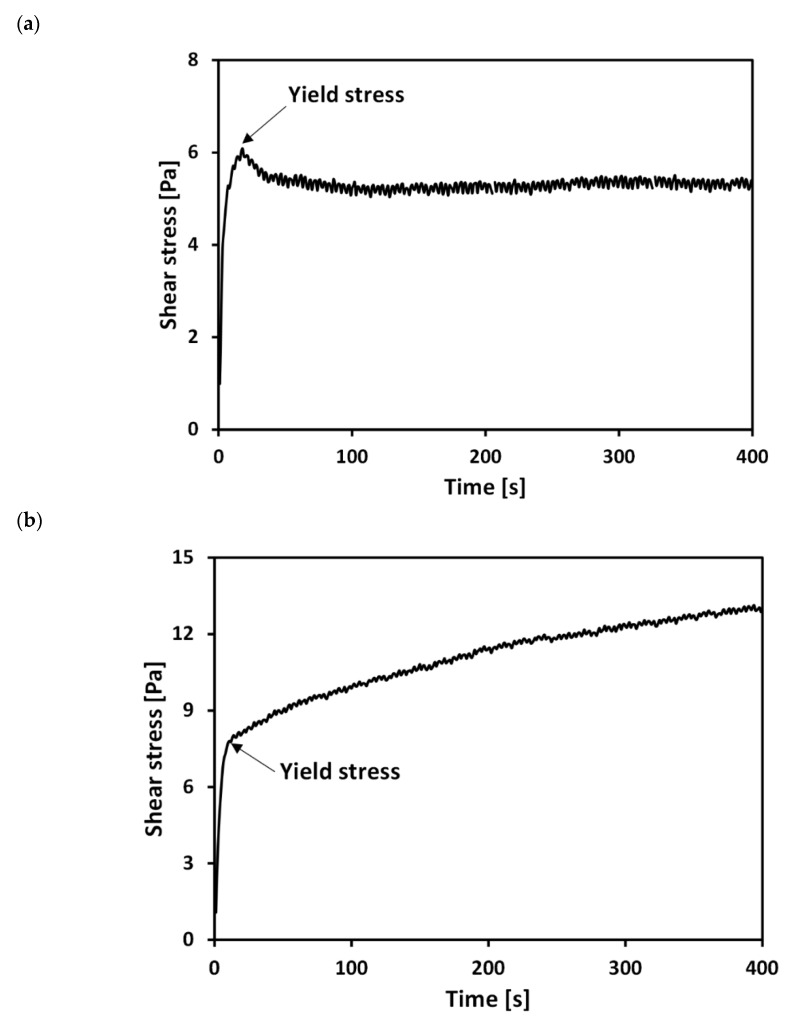
Typical stress–time curves of (**a**) untreated (water content = 218%) and (**b**) treated (water content = 263%) clays.

**Figure 4 materials-13-02525-f004:**
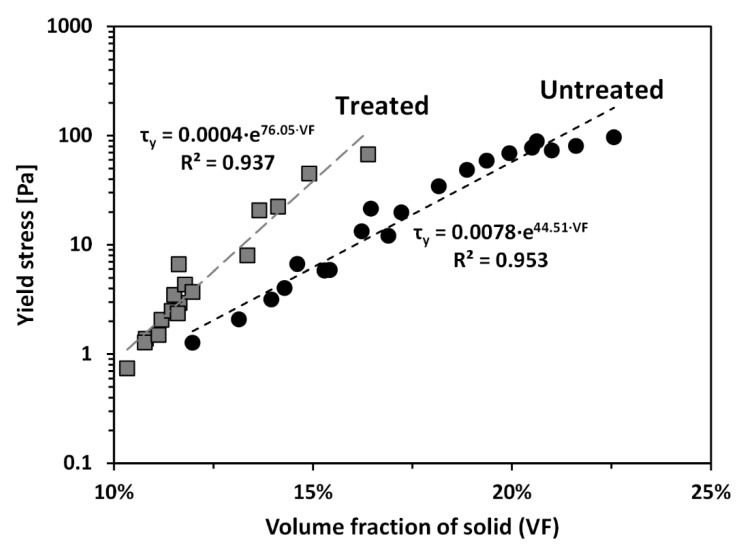
Yield stresses of untreated and treated clays as a function of volume fraction of solid (*VF*).

**Figure 5 materials-13-02525-f005:**
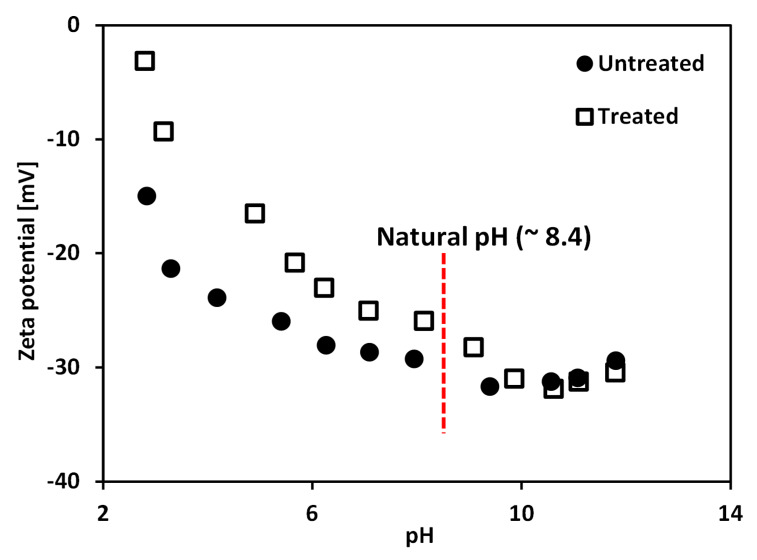
Variations of zeta potential of treated and untreated clays as a function of pH. Note zeta potential was measured using zeta potential analyzer (Otsuka Electronics, ELSZ-1000, Osaka, Japan) with the sample of 0.6 g of dry soil dispersed in 1 L of 0.01 M NaCl solution.

**Figure 6 materials-13-02525-f006:**
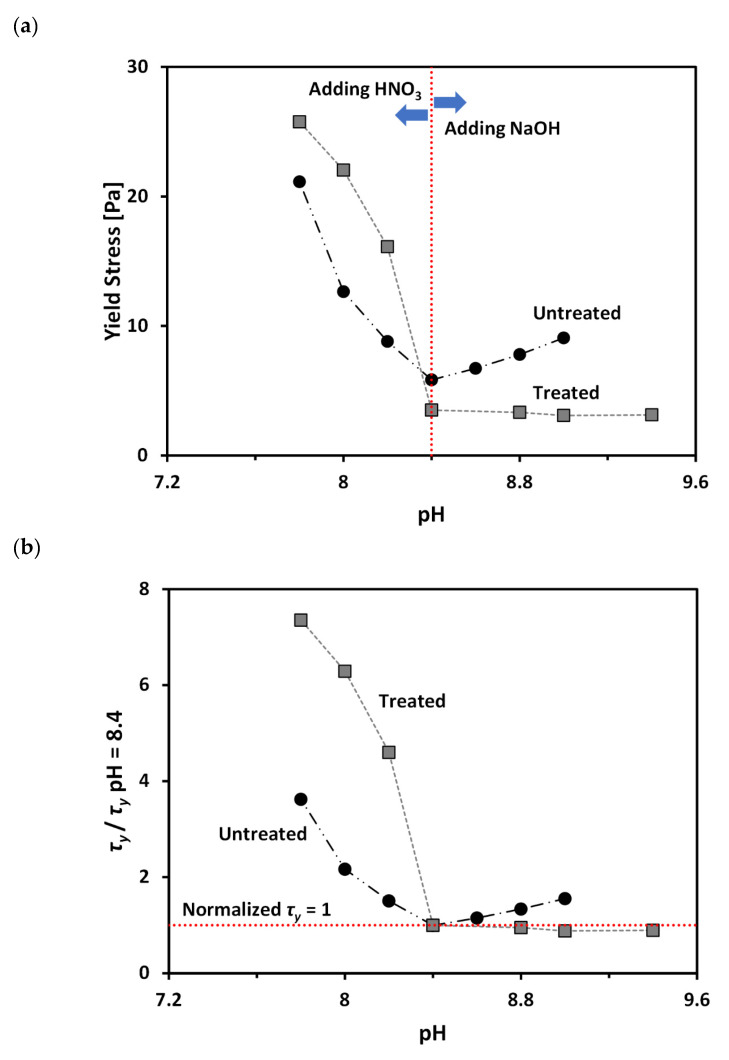
Effect of pH on the yield stress of treated and untreated clays: (**a**) variation of yield stress; and (**b**) variation of normalized yield stress (yield stress/yield stress at pH of 8.4).

**Table 1 materials-13-02525-t001:** Chemical composition of the calcium bentonite.

Component	SiO_2_	Al_2_O_3_	Fe_2_O_3_ ^a^	CaO	K_2_O	MgO	Na_2_O	TiO_2_	MnO	P_2_O_5_	LOI ^b^
wt.% composition	65.06	15.43	3.94	2.33	2.06	1.28	1.24	0.52	0.11	0.09	6.77

^a^ Fe_2_O_3_ = total Fe; ^b^ Loss of ignition

**Table 2 materials-13-02525-t002:** Index properties of the tested clays.

Sample	*G_s_*	LL [%]	PL [%]	*S_a_* [m^2^/g]	pH	CEC [cmol/kg]	USCS
Untreated	2.51	86.76	35.83	260.56	8.48	89.65	CH
Treated	2.47	121.53	38.10	263.62	8.46	89.21	CH

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
