# Peer review of "Effect of pH Variations on the Yield Stress of Calcium Bentonite Slurry Treated with pH-Responsive Polymer"

_materials, 2020, doi:10.3390/ma13112525_

Round 1

Reviewer 1 Report

The authors have studied the effect of pH variations on the yield stress of calcium bentonite both, untreated and treated with a pH-responsive polymer (polyacrylamide - PAM) by varying volume fractions of solid and pH conditions.

Some revisions are necessary:

1. For better view/follow, I recommend to the authors to put the references between right brackets.

2. At page 3, lines 86-90: For better understanding of the interaction between PAM molecules and clay particles, I recommend to the authors to insert a brief scheme.

3. At page 3, line 91, Figure 2: How low/high should be pH for these behaviours? It would be useful to insert some values.

4. At page 4, line 102: How long was dried the slurry in the oven at 110°C?

5. At page 6, lines 159-161: What instrument used the authors in order to measure Zeta potential?

6. At pages 6-7, lines 162-163: The authors mention that "At a pH of around 8.4, it can be observed in Figure 5 that the net ζ of treated clay is smaller than that of untreated clay...", but in Figure 5 we can see that this situation occurs at pH around 11, not 8.4. How is correct?

7. At page 7, line 177: I suggest to insert the location "...and untreated clays (Figure 4):...".

Reviewer 2 Report

Dear Editor,

dear authors,

In the present work an effort was made to evaluate the effect of pH changes on the yield stress of calcium bentonite treated with polyacrylamide. Towards the improvement of the manuscript I recommend the followings:

The citations throughout the text as well as the reference style are absolutely incorrect. Check the authors' guidelines of the journal carefully and proceed to the relative actions.

Abstract: Check for the text for syntax errors.

Experimental section:

- Give the X-Ray diffraction and the X-Ray fluorescence experimental parameters of the followed procedures in details.

- Page 2 (lines 66-68): "The measured specific gravity (Gs) was 2.51 (ASTM-D854). The plastic limit (PL) was 66 determined to be 35.83% by the traditional thread-rolling method (ASTM-D4318), and the liquid limit 67 (LL) was determined to be 86.76% by the fall cone method (BS-1377)."

It is not clear whether these parameters were measured by authors. If so, please provide a brief description of the associated techniques and discuss the determined values in the Results and discussion section.

- Page 2 (Figure 1): In the "y axis label" use arbitrary units.

- Page 3 (lines 80-90): There is much information which should be stated in the Results and discussion section.

- Page 3 (line 98): Please avoid the word "Note" when describing an experimental procedure.

- Page 4 (line 104): Avoid the repetition of the phrase "The index properties of the PAM treated clay are tabulated in Table 2."

Results and discussion section:

- Page 4 (line 124): Avoid the word "Note".

- Page 5-6 (lines 135-137): From the chemical point of view, the authors should be more concerned if it would be safe enough to claim that "the adsorption of the PAM on the clay surface can develop the additional bonds (or links) between clay particles". Adsorption mechanism due to Van der Waals forces should be examined in depth and an explanation should be given why it was not preferred.

-Page 7 (Figure 5): Please correct the word "Natrual".

Conclusions section: Please avoid the enumeration of your findings, and provide them in a more brief way.

Round 2

Reviewer 2 Report

Dear Editor,

dear Authors,

I can confirm that my suggestions were taken into consideration by authors and carefully transferred to the revised version of the manuscript.

Thank you for the cooperation.